# EFFICIENT CORRESPONDENCE LEARNING FOR DENSE SEMANTIC LABEL PROPAGATION

## ABSTRACT

Self-supervised learning (SSL) aims to learn robust and transferable representations purely from unlabeled data, which is especially useful when annotated data is scarce. Over the past decade, SSL has advanced significantly through paradigms such as Masked Image Modeling (MIM) and self-distillation. More recently, several methods have been designed for specific downstream tasks. In particular, SiamMAE introduced siamese masked auto-encoding for label propagation, where dense semantic labels from initial video frames are propagated to subsequent ones through inter-frame correspondence. CropMAE later showed that still images can achieve similar results by extracting two related crops (with random flipping to simulate a change of viewpoints between the two images) and reconstructing one from the other. While both methods are effective, they rely on reconstructing raw pixel values of masked patches, which cannot capture high-level semantics and is less robust than latent or semantic reconstruction. Building on insights from iBOT and DINOv2, we propose **Crop-CoRe**, an SSL method that extends CropMAE by reconstructing cluster assignments instead. In our experiments, Crop-CoRe consistently outperforms SiamMAE and CropMAE on label propagation benchmarks and achieves competitive results compared to state-of-the-art methods while requiring fewer training iterations. Moreover, it avoids reliance on video datasets or frame extraction, making it more resource-efficient. The code will be publicly released after publication.

## 1 INTRODUCTION

Self-supervised learning (SSL) has emerged as a promising paradigm for learning meaningful and robust representations from data without the need for human annotation. In particular, the absence of annotation reduces the bias towards a specific task, making the representations learned by SSL methods transferable to many downstream tasks. SSL has been successfully applied in many domains, ranging from natural language, images, videos, and audio. Practically, SSL is implemented by designing and solving a pretext task, which is a learning signal derived from the data itself. The most prominent methods in the image SSL literature are Contrastive Learning (CL) and Masked Image Modeling (MIM).

MIM is an SSL method that uses the pretext task of masking a portion of an image and learning to reconstruct the masked parts based on the visible ones. This idea draws inspiration from Masked Language Modeling (MLM), which was first introduced by BERT (Devlin et al., 2019) in the language domain. While performing masked modeling is straightforward in the language domain, since language can be easily parsed into discrete and semantic units, the continuous nature of images makes it more challenging to apply this paradigm. Some works, such as Masked Autoencoders (MAEs) (He et al., 2022), directly learn to reconstruct the pixel values of masked image patches. Although successful, reconstructing pixel values is a low-level task that does not produce high-level semantic features (Zhou et al., 2022). Bao et al. (2022) proposed a two-stage approach. A discrete variational autoencoder is first trained to tokenize images into discrete semantic units, and then MIM is performed in a second training stage by learning to predict the tokens of the masked image patches. Zhou et al. (2022) proposed to bootstrap these semantic units by learning an online tokenizer through self-distillation. Similar to DINO (Caron et al., 2021), they use a teacher-student architecture. Each network is composed of a ViT (Dosovitskiy et al., 2021) encoder and a clustering MLP head. The image patch sequence is masked in the student branch by replacing the masked

patches with a learnable mask token, and then the sequence is encoded and projected to token-wise softmax distributions. The same process is performed in the teacher branch on the unmasked patch sequence, and the task is to make the student match the teacher's distributions corresponding to the masked patches. This approach has been successful and is at the core of state-of-the-art SSL methods such as DINOv2 (Oquab et al., 2024).

Although SSL is usually intended to learn generalist features that transfer well to many downstream tasks, the way SSL methods are designed can bias them towards specific types of downstream tasks. For instance, image-level methods (Caron et al., 2021; Bardes et al., 2022; Caron et al., 2020; He et al., 2020; Chen et al., 2020) usually transfer better to image-level tasks such as image classification, while patch-level or dense methods (He et al., 2022; Bardes et al., 2022; Zhou et al., 2022; Oquab et al., 2024) usually transfer better to dense downstream tasks such as semantic segmentation. In this work, we are particularly interested in designing an SSL method that transfers well to dense semantic label propagation tasks. SiamMAE (Gupta et al., 2023) is a recently introduced method for these tasks that learns to reconstruct the pixels of masked patches of an image by using unmasked patches of another image as a reference. Hence, this reconstruction task is performed by leveraging the dense correspondence between patches of the two images. SiamMAE uses frames of a video as its reference and target images. CropMAE (Eymaël et al., 2024) introduced a more efficient way to implement this paradigm. Different crops of the same image with additional random horizontal flipping are used in place of video frames, alleviating the need for a video dataset and showing that this paradigm does not learn temporal features, such as motion, but inter-image correspondence. Although successful, both methods rely on directly reconstructing the pixel values of masked patches. Subsequently, T-CoRe (Liu et al., 2025) introduced a method similar to SiamMAE, but with two major differences. First, they reconstruct the cluster assignments of the masked patches, like in iBOT (Zhou et al., 2022) and DINOv2 (Oquab et al., 2024). Second, they reconstruct patches of a frame in a "sandwich sampling" fashion by using both a past and a future frame as references, with the intuition to reduce the uncertainty of reconstructing a present frame from a past frame. We argue that, despite this strategy, the uncertainty remains. Moreover, learning dense correspondence does not require a video dataset, as demonstrated by (Eymaël et al., 2024).

Based on these observations, we introduce **Crop-CoRe**, a **Crop CoRe**spondence learning method that extends CropMAE by reconstructing cluster assignments of masked patches, with the student trained to match the teacher's prototype assignments rather than raw pixels. Following CropMAE, we adopt a cropping strategy that removes uncertainty and makes the task deterministic, and eliminates the need for a video dataset. Crop-CoRe outperforms both SiamMAE and CropMAE on 3 label propagation benchmarks and achieves competitive results compared to T-CoRe and other state-of-the-art methods while requiring significantly fewer training iterations. We summarize our contributions as follows:

- We propose **Crop-CoRe**, a new SSL method for downstream dense semantic label propagation tasks. Our method achieves competitive results compared to state-of-the-art methods.

- We show that **Crop-CoRe** is effective and efficient, requiring no video dataset and converging faster thanks to the deterministic design of its pretext task.

- We provide further evidence for the effectiveness of latent-space reconstruction by showing that predicting cluster assignments yields better semantic features than CropMAE's pixel-space reconstruction, making it more consistent with the goal of propagating dense semantic labels.

## 2 RELATED WORKS

### 2.1 SELF-SUPERVISED IMAGE REPRESENTATION LEARNING

Self-supervised learning methods can be broadly categorized into two main types: contrastive and non-contrastive methods.

*Contrastive methods* Hadsell et al. (2006); Oord et al. (2018); Hjelm et al. (2019); Bachman et al. (2019); Wu et al. (2018); He et al. (2020); Chen et al. (2020) train a network to give similar embeddings to samples sharing the same semantics (positives), or dissimilar embeddings otherwise (negatives). Negative samples are primarily needed to avoid representation collapse. Initial methods

Hadsell et al. (2006); Oord et al. (2018); Hjelm et al. (2019); Bachman et al. (2019) use in-batch samples as negatives, making it challenging to have many negatives when computational resources limit large batch sizes. Wu et al. (2018) proposed memory banks to untie the number of negatives from the batch size, allowing support for larger numbers of negatives. However, this approach Wu et al. (2018) requires storing a representation of all images in the dataset inside the memory bank, which is not scalable for large datasets. He et al. (2020) introduced the momentum encoder to circumvent this limitation. The momentum encoder has an identical architecture to the main encoder. The latter is updated with backpropagation, while the former is an exponential moving average of the latter. Chen et al. (2020) proposed a simplified framework compared to prior methods and introduced several good practices that have since been widely adopted in the SSL literature.

*Non-contrastive methods* learn a representation without using negative samples. Regularized methods Zbontar et al. (2021); Bardes et al. (2022) use explicit regularization terms to avoid collapse. Clustering-based methods Caron et al. (2018; 2020; 2021) train a network by bootstrapping an abstract clustering of samples during training. DeepCluster (Caron et al., 2018) alternates k-means clustering and supervised training with the obtained pseudo-labels. This approach, however, requires computing features of all samples in the training set every time before the k-means clustering, which limits its scalability to larger datasets. SwAV (Caron et al., 2020) introduced an online clustering method that learns cluster prototypes thanks to a swapped cluster prediction mechanism between two augmented versions of the same image. Specifically, each predicts the cluster assignment of the other. Subsequently, Caron et al. (2021) introduced DINO, a clustering method that bootstraps cluster prototypes thanks to a teacher-student architecture. Similar to MoCo (He et al., 2020), the teacher and student have identical architectures, the student being updated with backpropagation, while the teacher is an exponential moving average of the student. The representation collapse is avoided through a centering and sharpening strategy of the teacher's output distribution. **Crop-CoRe** is a clustering-based method. More precisely, Crop-CoRe performs clustering at both the image level and the patch level.

## 2.2 MASKED IMAGE MODELING (MIM)

Since the introduction of BERT (Devlin et al., 2019), masked modeling has gained significant traction in the field of SSL. Masked modeling is a pretext task that consists of masking some parts of the input data and learning to reconstruct the masked parts based on the visible ones. Inspired by the pioneering work of BERT in the language domain, many efforts have been made to apply this paradigm in the vision domain. One critical challenge is that the continuous nature of images makes it difficult to apply masked modeling, in contrast to language, which is discrete. BEiT (Bao et al., 2022) was the first work to adopt this paradigm for images and addressed this continuity challenge by learning a discrete representation of images first. Their method consists of two stages. The first consists of training a discrete variational autoencoder (dVAE) Ramesh et al. (2021) to tokenize an image into discrete tokens, and the second involves training an encoder to reconstruct masked image patches in a BERT-like fashion. Subsequently, He et al. (2022) introduced the MAE architecture, an asymmetric encoder-decoder design in which the encoder only sees visible patches and the decoder is lightweight compared to the encoder, reconstructing the pixels inside the masked patches. This design choice significantly reduces the computation costs and demonstrates that we can successfully perform MIM by directly reconstructing pixels. Hence, it removes the need to train an image tokenizer beforehand. Zhou et al. (2022) highlights that such a paradigm, however, struggles in semantic abstraction. For instance, since images are not semantically dense, a masked patch can be easily reconstructed by looking at nearby visible ones without leveraging any semantic knowledge.

On the other hand, the success of masked language modeling has been primarily attributed to the ability to tokenize text into semantically meaningful pieces. Although training a tokenizer before MIM (Bao et al., 2022) is a successful approach, this requires training a dVAE offline. This may not generalize well to different architectures and data from different domains. Hence, Bao et al. (2022) introduced iBOT, a method that learns this tokenization online through a DINO-like self-distillation. DINOv2 Oquab et al. (2024) built on iBOT, providing techniques to scale the size of the model and the amount of data while maintaining stability. **Crop-CoRe** follows this line of work by reconstructing tokens of an online tokenizer.

## 2.3 CORRESPONDENCE LEARNING

Correspondence learning aims to learn how to associate pixels of two images that feature different views of the same scene, such as frames of a video. One important application is video propagation tasks, such as semi-supervised video object segmentation, which aims to propagate the segmentation of initial video frames to subsequent ones. SiamMAE (Gupta et al., 2023) demonstrated a state-of-the-art performance on video propagation tasks Pont-Tuset et al. (2017); Jhuang et al. (2013); Zhou et al. (2018). Building on MAE, SiamMAE is a siamese masked modeling paradigm in which two frames of a video are asymmetrically masked ($0\%$ and $95\%$), and a cross-attention decoder is used to reconstruct pixels of the masked frame by "looking" at the unmasked frame. Perfectly implementing the idea of propagating information from one frame to another. Subsequently, CropMAE (Eymaël et al., 2024) builds on SiamMAE and use different crops of the same image in place of video frames. Their approach achieves very competitive performance compared to SiamMAE and is significantly more efficient in terms of memory consumption and training convergence. However, both of these methods directly reconstruct pixels. This has been known to capture low-level semantic features (Zhou et al., 2022). Building on DINOv2, T-CoRe (Liu et al., 2025) propose a similar approach to SiamMAE but learns to reconstruct the cluster assignments of the masked patches. In this work, we propose a method inspired by CropMAE and T-CoRe. Specifically, $(i)$ we leverage an image dataset and extract different crops from still images, and $(ii)$ we learn to reconstruct the cluster assignments of masked patches.

## 3 METHOD

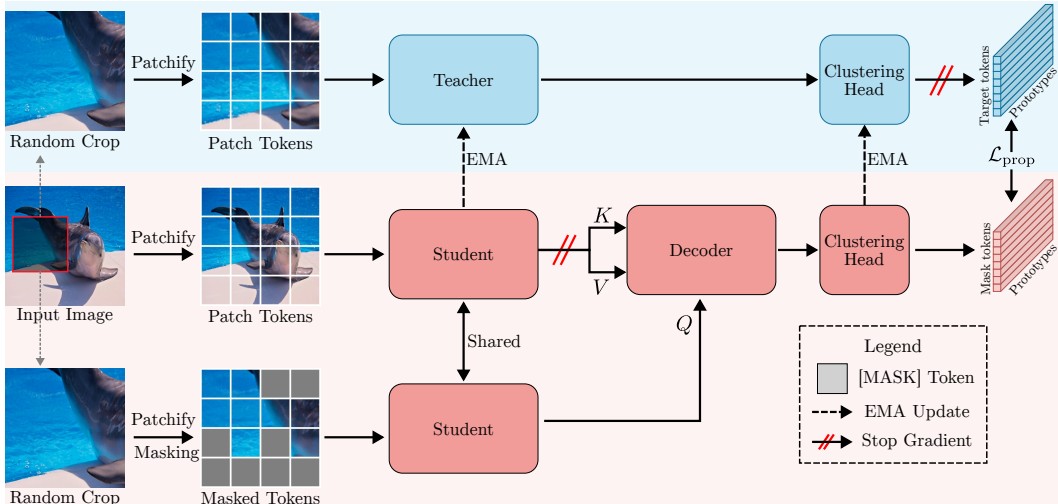

Figure 1: **Overview of Crop-CoRe.** A first global crop is extracted from the input image. A second local crop is subsequently extracted from the global crop. The student network is tasked to reconstruct the masked local crop by referring to the global crop. Its targets are built by passing the unmasked local crop in the teacher branch. The objective is to reconstruct the soft cluster assignments of the masked patches.

### 3.1 PROPAGATION WITH SIAMESE MASKED AUTO-ENCODING

Figure 1 gives an overview of our method. The lower block represents the student branch (3.1.1), while the upper block represents the teacher branch (3.1.2). Next, we will describe each branch in detail.

#### 3.1.1 STUDENT BRANCH

The student branch is where the correspondence learning is performed. Following Crop-MAE (Eymaël et al., 2024), we sample two random views $\boldsymbol{V}_1, \boldsymbol{V}_2 \in \mathbb{R}^{H \times W \times 3}$ from an image $\boldsymbol{I}$

by random cropping, resizing, and horizontal flipping, where $H$ and $W$ are the height and width of the views. Each view is then divided into a sequence $\boldsymbol{v}_i = \{\boldsymbol{v}_i^j\}_{j=1}^N$ of $N = \frac{HW}{p^2}$ patches containing $p^2$ pixels each. The sequence $\boldsymbol{v}_2$ is masked to form another sequence $\bar{\boldsymbol{v}}_2 = \{\bar{\boldsymbol{v}}_2^j\}_{j=1}^N$ where $\bar{\boldsymbol{v}}_2^j = (1 - \boldsymbol{m}_j) \cdot \boldsymbol{v}_2^j + \boldsymbol{m}_j \cdot [\text{MASK}]$, where $\boldsymbol{m} \in \{0, 1\}^N$ is a vector indicating whether a patch is masked and [MASK] is a learnable token used to reconstruct masked patches, following Oquab et al. (2024). Each sequence is further prepended with a [CLS] token, another learnable token that will be used for image-level representation learning.

A ViT (Dosovitskiy et al., 2021) encoder $f_s$ is used to encode $\boldsymbol{v}_1$ and $\bar{\boldsymbol{v}}_2$ into latent representations $\boldsymbol{z}_1$ and $\bar{\boldsymbol{z}}_2 \in \mathbb{R}^{N \times D}$ where $D$ is the latent dimensionality after the encoder. Subsequently, the contextualized masked tokens of $\bar{\boldsymbol{z}}_2$ are processed by a cross-attention decoder $g$ to compute the propagated latent representation $\bar{\boldsymbol{z}}_2^p = g(\bar{\boldsymbol{z}}_2[\boldsymbol{m}], \boldsymbol{z}_1)$, where $\bar{\boldsymbol{z}}_2[\boldsymbol{m}]$ is the set of contextualized masked tokens, that is, the tokens corresponding to the masked patches. Hence, similarly to Crop-MAE, we use an asymmetric masking strategy with an encoder-decoder to propagate information from one crop to another. One key difference is that our encoder sees the visible and masked tokens, while the decoder only sees the masked tokens. This scheme is reversed in CropMAE. Additionally, rather than reconstructing the pixels themselves, we reconstruct their cluster assignments thanks to self-distillation (Caron et al., 2021), as described in the next sections.

### 3.1.2 TEACHER BRANCH

The teacher encoder $f_t$ has an identical architecture to the student encoder $f_s$. In the teacher branch (upper block in fig. 1), we build the targets for the masked tokens in the student branch. To achieve that, the representations of the unmasked second view $\boldsymbol{v}_2$ are computed by the teacher encoder as $\boldsymbol{z}_2^t = f_t(\boldsymbol{v}_2)$, and the outputs corresponding to the masked patches $\boldsymbol{z}_2^t[\boldsymbol{m}]$ are used to form the targets in the student branch. Note that the teacher needs to be fed with the view that is being reconstructed, $\boldsymbol{v}_2$ here precisely, in order to establish a position-wise correspondence between the predictions of the student and the targets created by the teacher.

### 3.2 RECONSTRUCTION WITH SELF-DISTILLATION

While prior works Gupta et al. (2023); Eymaël et al. (2024) directly reconstruct the pixels, we follow the approach of recent works Zhou et al. (2022); Oquab et al. (2024); Liu et al. (2025) to predict the cluster assignments of the masked patches. Intuitively, this is closer to the initial idea of semantic label propagation between frames since cluster assignments can be thought of as latent semantic labels.

Formally, the student clustering head $h_s$ and the teacher clustering head $h_t$, respectively use $\bar{\boldsymbol{z}}_2^p$ and $\boldsymbol{z}_2^t[\boldsymbol{m}]$ to compute the distribution of the $i$-th masked patch, and corresponding visible patch as follows:

$$P_s(\bar{\boldsymbol{v}}_2[\boldsymbol{m}])_i^{(j)} = \frac{\exp(h_s(\bar{\boldsymbol{z}}_2^p)_i^{(j)}/\tau_s)}{\sum_{k=1}^K \exp(h_s(\bar{\boldsymbol{z}}_2^p)_i^{(k)}/\tau_s)}, \tag{1}$$

$$P_t(\boldsymbol{v}_2[\boldsymbol{m}])_i^{(j)} = \frac{\exp(h_t(\boldsymbol{z}_2^t[\boldsymbol{m}])_i^{(j)}/\tau_t)}{\sum_{k=1}^K \exp(h_t(\boldsymbol{z}_2^t[\boldsymbol{m}])_i^{(k)}/\tau_t)}, \tag{2}$$

where $\tau_s$ and $\tau_t$ respectively their temperature parameters. To avoid representation collapse, a sharpening and centering is applied to the teacher's output distribution (Caron et al., 2021).

Finally, the propagation loss writes:

$$\mathcal{L}_{\text{prop}} = \frac{1}{|\boldsymbol{m}|} \sum_{i=1}^{|\boldsymbol{m}|} H(P_t(\boldsymbol{v}_2[\boldsymbol{m}])_i, P_s(\bar{\boldsymbol{v}}_2[\boldsymbol{m}])_i) = -\frac{1}{|\boldsymbol{m}|} \sum_{i=1}^{|\boldsymbol{m}|} \sum_{j=1}^K P_t(\boldsymbol{v}_2[\boldsymbol{m}])_i^{(j)} \log(P_s(\bar{\boldsymbol{v}}_2[\boldsymbol{m}])_i^{(j)}),$$
$$\tag{3}$$

where $H$ is the cross-entropy function, $|\boldsymbol{m}|$ is the number of non-zero elements of $\boldsymbol{m}$.

### 3.3 IMAGE-LEVEL REPRESENTATION LEARNING

In addition, following DINOv2 (Oquab et al., 2024) and T-CoRe (Liu et al., 2025), we use the [CLS] token representations to perform image-level representation learning. Precisely, the second view $v_2$ and 8 other small crops extracted from $I$ are used to compute $\mathcal{L}_{\text{DINO}}$ loss and the KoLeo regularizer is used to promote a uniform span of the features within a batch with a $\mathcal{L}_{\text{koleo}}$ term. Refer to the related works for more information on these terms.

### 3.4 TRAINING CROP-CORE

To summarize, **Crop-CoRe** is trained with the following loss:

$$\mathcal{L} = \mathcal{L}_{\text{DINO}} + \lambda_1 \mathcal{L}_{\text{prop}} + \lambda_2 \mathcal{L}_{\text{koleo}} \,, \tag{4}$$

where $\lambda$ is the strength of the KoLeo regularizer term. During training, the student is updated with backpropagation, and the teacher is updated as an exponential moving average of the student:

$$\theta_t \leftarrow m \cdot \theta_t + (1 - m) \cdot \theta_s \,, \tag{5}$$

where $\theta_s$ and $\theta_t$ are the parameters of the student and the teacher, and $m$ is the momentum coefficient.

## 4 EXPERIMENTS

### 4.1 IMPLEMENTATION DETAILS

**Pre-training.** In all our experiments, we use ViT-S/16 (Dosovitskiy et al., 2020) as the encoder. Following Liu et al. (2025), our decoder is a one-layer cross-attention decoder. An individual layer is composed of a self-attention, a cross-attention mechanism, and an MLP. For pre-training, we mainly use the ImageNet-1k dataset (Russakovsky et al., 2015). The terms *global crop* and *local crop* usually refer, in the SSL literature, to $224 \times 224$ and $94 \times 94$ resolution crops, respectively. To avoid confusion, we will refer to these as *high-resolution* and *low-resolution* crops. Hence, the term *global crop* will refer to an anchor *high-resolution crop*, and any *high-resolution crop* extracted from this anchor will be referred to as a *local crop* with respect to this anchor. In each training iteration, 2 high-resolution crops are extracted from the original image, and then from each, a local crop is extracted to form a reference-target pair. Subsequently, 8 low-resolution crops are also extracted from the original image. Following (Oquab et al., 2024), $50\%$ of the local crops are masked in the student branch with a uniformly sampled masking ratio in $[0.1, 0.5]$. The global crops and their corresponding local crops are used in the propagation part of our method, and the 2 local crops and 8 low-resolution crops are used for the image-level representation learning part of our method with $\mathcal{L}_{\text{DINO}}$. Additional details on pre-training and evaluation settings are provided in the Appendix.

**Optimization.** During training, the ViT-S/16 is trained for 50 epochs with an effective batch size of 1024 distributed between 4 GPUs. The student is optimized with the AdamW (Loshchilov & Hutter, 2019) optimizer. Following Liu et al. (2025), the learning rate for the student branch $lr$ is set to $1 \times 10^{-3}$ and decays to $1 \times 10^{-6}$ with a cosine schedule. For the decoder, the learning rate is set to $0.1 \times lr$. The weights of the loss function are set to $\lambda_1 = 0.8$ and $\lambda_2 = 0.1$ following Liu et al. (2025).

**Data augmentations.** Besides the multiple crops (Caron et al., 2020), we apply random Gaussian blur, grayscale, color jittering, and horizontal flips to the crops following previous works in the literature(Chen et al., 2020; Oquab et al., 2024).

### 4.2 MAIN RESULTS AND DISCUSSIONS

We compare our method with the state-of-the-art methods in three downstream tasks: semi-supervised video object segmentation on DAVIS (Pont-Tuset et al., 2017), semantic part propagation on VIP (Zhou et al., 2018) and pose keypoint propagation on JHMDB (Jhuang et al., 2013). The results are reported in table 1. The following observations can be made: **1)** Crop-CoRe outperforms Crop-MAE on all benchmarks, showing the benefit of performing the reconstruction task in a latent space. This is further illustrated in fig. 2, in which we can notice that CropMAE is very sensitive to

| Type | | Method | Backbone | Dataset | Epoch | DAVIS-2017 | | | VIP | JHMDB | |
|------|--|--------|----------|---------|-------|------------|--|--|-----|-------|--|
| | | | | | | $\mathcal{J}\&\mathcal{F}_\mathrm{m}$ | $\mathcal{J}_\mathrm{m}$ | $\mathcal{F}_\mathrm{m}$ | mIoU | PCK@0.1 | PCK@0.2 |
| Image-level SSL | | SimCLR[†] ICML'20 | ViT-S/16 (22M) | Kinetics-400 | 400 | 53.9 | 51.7 | 56.2 | 31.9 | 37.9 | 66.1 |
| | | Moco v3[†] ICCV'21 | ViT-S/16 (22M) | Kinetics-400 | 400 | 57.7 | 54.6 | 60.8 | 32.4 | 38.4 | 67.6 |
| | | DINO[†] ICCV'21 | ViT-S/16 (22M) | ImageNet-1k | 800 | 61.8 | 60.2 | 63.4 | 36.2 | 45.6 | 75.0 |
| | | DINO[†] ICCV'21 | ViT-S/16 (22M) | Kinetics-400 | 400 | 59.5 | 56.5 | 62.5 | 33.4 | 41.1 | 70.3 |
| | | ODIN²[†] ECCV'22 | ResNet50 (26M) | ImageNet-1k | 1000 | 54.1 | 54.3 | 53.9 | / | / | / |
| | | CrOC[†] CVPR'23 | ViT-S/16 (22M) | ImageNet-1k | 300 | 44.7 | 43.5 | 45.9 | 26.1 | / | / |
| Masked Modeling | Pixel Space | MAE[†] CVPR'22 | ViT-B/16 (87M) | ImageNet-1k | 1600 | 53.5 | 52.1 | 55.0 | 28.1 | 44.6 | 73.4 |
| | | RC-MAE[†] ICLR'23 | ViT-S/16 (22M) | ImageNet-1k | 1600 | 49.2 | 48.9 | 50.5 | 29.7 | 43.2 | 72.3 |
| | | SiamMAE[†] NeurIPS'23 | ViT-S/16 (22M) | Kinetics-400 | 400 | 57.6 | 56.0 | 60.0 | 33.2 | 46.1 | 74.0 |
| | | SiamMAE[†] NeurIPS'23 | ViT-S/16 (22M) | Kinetics-400 | 2000 | 62.0 | 60.3 | 63.7 | 37.3 | 47.0 | 76.1 |
| | | CropMAE[†] ECCV'24 | ViT-S/16 (22M) | ImageNet-1k | 400 | 60.4 | 57.6 | 63.3 | 33.3 | 43.6 | 72.0 |
| | | RSP[†] ICML'24 | ViT-S/16 (22M) | Kinetics-400 | 400 | 60.1 | 57.4 | 62.8 | 33.8 | 44.6 | 73.4 |
| | | CDG-MAE-a1[†] Arxiv'25 | ViT-S/16 (22M) | ImageNet-1k | 100 | 61.2 | 57.4 | 64.3 | 37.6 | 46.5 | 75.5 |
| | | CDG-MAE-a3[†] Arxiv'25 | ViT-S/16 (22M) | ImageNet-1k | 100 | 62.6 | 59.7 | 65.5 | 38.1 | 47.8 | 76.3 |
| | Latent Space | iBOT[‡] ICLR'22 | ViT-S/16 (22M) | ImageNet-1k | 800 | 62.6 | 60.2 | 65.1 | 38.0 | 44.3 | 74.4 |
| | | DINO v2[†] TMLR'24 | ViT-S/16 (22M) | ImageNet-22k | 100 | 63.2 | 61.4 | 65.1 | 37.3 | 46.3 | 75.4 |
| | | T-CoRe[‡] CVPR'25 | ViT-S/16 (22M) | ImageNet-1k | 100 | 64.1 | 62.1 | 66.1 | 39.6 | 46.2 | 75.5 |
| | | T-CoRe[‡] CVPR'25 | ViT-S/16 (22M) | Kinetics-400 | 400 | 64.8 | 63.5 | 66.0 | 37.9 | 46.9 | 75.2 |
| | | **Crop-CoRe (Ours)** | ViT-S/16 (22M) | ImageNet-1k | 50 | 64.9 | 62.6 | 67.1 | 37.5 | 44.9 | 74.3 |

Table 1: **Main results.** Comparison with prior methods on three dense-level video downstream tasks. Results on baselines directly reported from previous studies. Missing values represent the absence of reported results or implementations. The best and second-best results are highlighted in soft red and soft blue, respectively.

pixel intensity variations. This is expected since the pretext task of CropMAE aims to reconstruct the exact pixel values. Reconstructing the cluster assignments of patches is more closely aligned with the downstream task of semantic label propagation, making our method more robust to illumination variations. **2)** Crop-CoRe achieves the best average score on DAVIS, even surpassing the version of T-CoRe pre-trained on Kinetics-400 (Kay et al., 2017), showing the effectiveness of our method on semi-supervised video object segmentation. **3)** Crop-CoRe slightly underperforms on pose key-point and semantic part propagation compared to state-of-the-art methods. The gap is particularly more pronounced for methods pre-trained on Kinetics-400. This observation is consistent with the experiments of (Belagali et al., 2025). The main reason is the limited change of viewpoints between the global and local crops that is inherently present between frames of a video. To adapt T-CoRe to image datasets, Liu et al. (2025) simulates the relative changes between video frames by using $k$-NN images as references for each target image. Belagali et al. (2025) uses a diffusion model (Belagali et al., 2024) to create, for each image, a bag of views with varied changes in motion, perspective, and pose. This results in improvements on VIP and JHMDB for both methods. However, their approaches require offline preprocessing, which is an additional overhead. In contrast, our method does not require any preprocessing. **4)** Crop-CoRe achieves competitive results on all benchmarks while requiring significantly fewer training iterations. We hypothesize that this is primarily due to the deterministic nature of our Global-to-Local reconstruction paradigm, which enables our method to learn faster.

## 4.3 Performance Analysis

In this section, we investigate the impact of different design choices in our method, including the number of training epochs, cropping strategy, number of prototypes, and masking strategy.

**Impact of training duration.** We tested different numbers of training epochs: 25, 50, 100, and 200. The results are reported in table 2a. We observe that Crop-CoRe achieves high performance on DAVIS quickly, reaching its peak after only 25 epochs, and maintains this performance at 50 epochs. However, in our experiments, we observed a decrease in performance after more epochs, as shown by the performance after 100 epochs. This is most likely due to a collapse of our dense features. This behavior has also been observed with CropMAE. Using a Gram loss (Siméoni et al., 2025) could solve it and make our method more scalable to longer training.

**Impact of the number of prototypes.** We tested different numbers of prototypes to see their influence on the overall performance. Table 2b shows that increasing the number of prototypes generally improves the performance. Intuitively, since the student network would have to match the teacher's distribution over a higher set of latent classes, this makes the task more challenging.

**Impact of the cropping strategy.** Following Eymaël et al. (2024), we tested Crop-CoRe on different cropping strategies: Global-to-Local, Local-to-Global and Random. Table 2c shows that the best

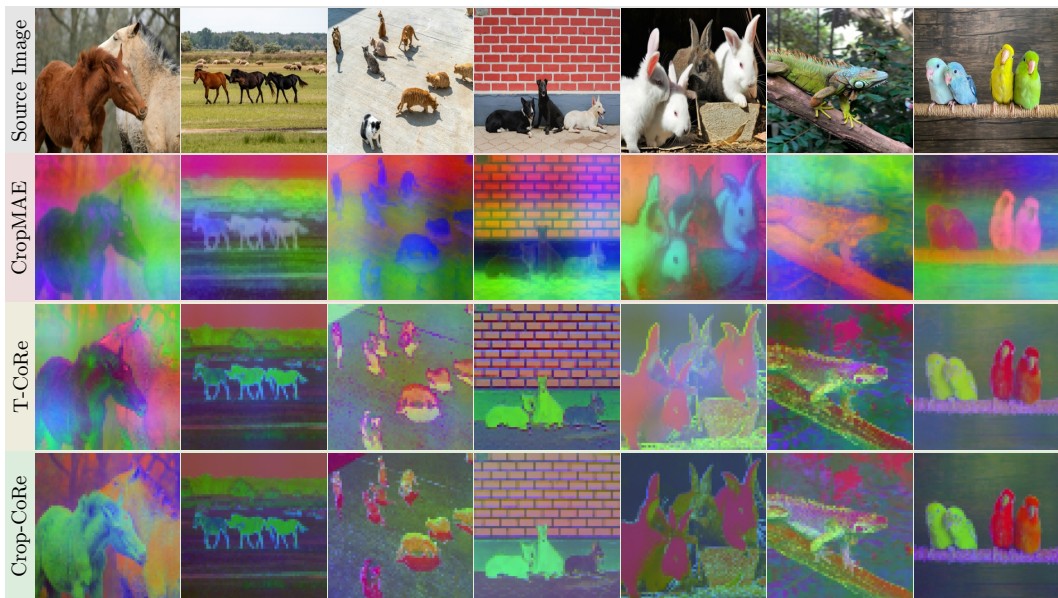

Figure 2: **PCA visualization.** We analyse with PCA the dense features of CropMAE (Eymaël et al., 2024), T-CoRe (Liu et al., 2025) and Crop-Core from top to bottom. We first notice that CropMAE is very sensitive to pixel intensity variations in contrast to T-CoRe and Crop-CoRe, which are more semantic- and instance-oriented. We can also notice that Crop-CoRe has similar results to T-CoRe, further emphasizing the non-necessity of a video dataset for correspondence learning.

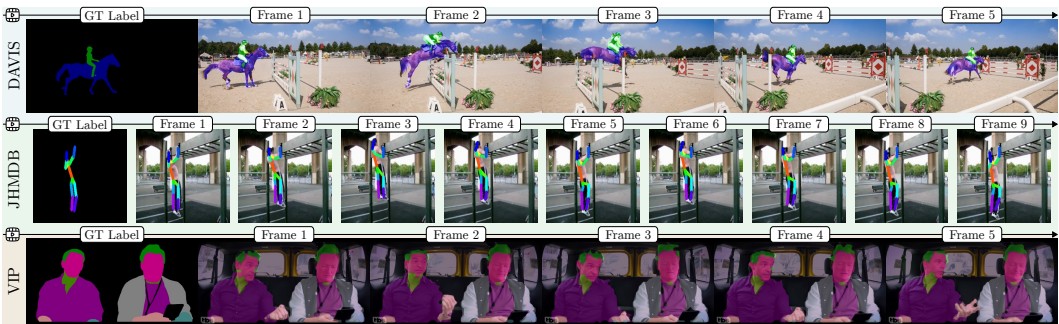

Figure 3: **Frame propagation.** We show how our method is able to correctly propagate the labeled information contained in the first labeled frame in subsequent frames.

performance is achieved with Global-to-Local reconstruction. Local-to-Global reconstruction is highly uncertain, as the model would have to reconstruct parts of the global crop that are not visible in the local crop, leaving many possibilities and ultimately requiring much longer training. Random cropping aims to use independently cropped images as a reference and a target. This can lead to scenarios where the two crops are completely unrelated. This makes the reconstruction task even more uncertain. The Global-to-Local reconstruction enables a consistent relationship between the reference and the target image, as well as a deterministic reconstruction task that is quickly learnable by the model. Overall, these observations are consistent with Eymaël et al. (2024).

**Impact of the masking scheme**. Different masking schemes have been proposed in the MIM literature, with random masking being the traditional and most widely used scheme. We additionally tested block masking, inverse block masking, cyclic masking (Darcet et al., 2025), and color masking (Hinojosa et al., 2024). As shown in table 2d, the random masking scheme yields the best performance. Interestingly, this contrasts with MIM methods, which do not reconstruct one image from another. While reconstructing a continuous block from nothing is more challenging, it is less chal-

| Epochs | DAVIS-2017 | | |
|---|---|---|---|
| | $\mathcal{J}\&\mathcal{F}_\mathrm{m}$ | $\mathcal{J}_\mathrm{m}$ | $\mathcal{F}_\mathrm{m}$ |
| 25 | 63.8 | 61.4 | 66.2 |
| 50 | **64.9** | **62.6** | **67.1** |
| 100 | 63.9 | 61.9 | 65.9 |

(a) Analysis on the number of epochs.

| Crop Strategy | DAVIS-2017 | | |
|---|---|---|---|
| | $\mathcal{J}\&\mathcal{F}_\mathrm{m}$ | $\mathcal{J}_\mathrm{m}$ | $\mathcal{F}_\mathrm{m}$ |
| Local-to-GLobal | 64.6 | 62.3 | 66.9 |
| Global-to-Local | **64.9** | **62.6** | **67.1** |
| Random | 64.1 | 61.9 | 66.4 |

(c) Analysis on the cropping strategy.

| K | DAVIS-2017 | | |
|---|---|---|---|
| | $\mathcal{J}\&\mathcal{F}_\mathrm{m}$ | $\mathcal{J}_\mathrm{m}$ | $\mathcal{F}_\mathrm{m}$ |
| 8192 | 62.5 | 60.2 | 64.8 |
| 16384 | 61.9 | 59.6 | 64.2 |
| 32768 | 63.2 | 60.8 | 65.7 |
| 65536 | **64.9** | **62.6** | **67.1** |

(b) Analysis on the number of prototypes.

| Mask Strategy | DAVIS-2017 | | |
|---|---|---|---|
| | $\mathcal{J}\&\mathcal{F}_\mathrm{m}$ | $\mathcal{J}_\mathrm{m}$ | $\mathcal{F}_\mathrm{m}$ |
| Random | **64.9** | **62.6** | **67.1** |
| Block | 62.4 | 59.9 | 64.9 |
| Inverse | 62.2 | 59.6 | 64.9 |
| CyclicMask | 63.2 | 60.6 | 65.8 |
| Red-Masking | 62.6 | 60.1 | 65.2 |
| Blue-Masking | 62.5 | 59.8 | 65.2 |
| Green-Masking | 63.7 | 61.3 | 66.2 |
| Purple-Masking | 63.1 | 60.6 | 65.6 |

(d) Analysis on the masking strategy.

Table 2: **Performance analysis.** We evaluate our method with different settings on DAVIS-2017. Default settings are highlighted in green . The best results are marked with **bold**. Table 2a shows that **Crop-CoRe** learns quickly, achieving very good results after only 25 training epochs. Table 2b shows that we generally improve results with more prototypes. Table 2c shows that the Global-to-Local reconstruction is optimal compared to Local-to-Global and Random. Table 2d shows random masking performs the best in a Global-to-Local reconstruction setting.

lenging to reconstruct in a Global-to-Local scenario. The same observation holds for inverse block masking. Cyclic masking introduces a level of randomness to the inverse block masking scheme, resulting in an improvement compared to the other two. We also tested recently introduced masking schemes, called ColorMasking (Hinojosa et al., 2024), for their improvements in pixel space reconstruction. However, these masking schemes did not improve our results. Overall, the random masking scheme ensures the highest complexity and, therefore, the best downstream performance.

## 5 CONCLUSION

In this work, we introduce Crop-CoRe, a self-supervised learning method targeted at downstream video label propagation tasks. Our experiments validate the effectiveness of our method. In particular, the Global-to-Local cropping strategy enables Crop-CoRe to achieve competitive performances while requiring significantly fewer training iterations compared to many baselines. Moreover, by outperforming CropMAE, we further support the idea of performing the reconstruction task in a latent space rather than in pixel space. Additionally, Crop-CoRe alleviates the need for video datasets, that are more costly to use for training, further demonstrating its efficiency. Further analyzing the behavior of our method, compared to a method pre-trained on a video dataset is a interesting venue for future work.

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

# A APPENDIX

## A.1 FURTHER TRAINING DETAILS

Table A.1 gives further details on our pre-training hyperparameters and network architecture.

| Hyperparameter | Notation | Value |
|---|---|---|
| Sampling strategy | | |
| Reference image | $v_1$ | Global crop |
| Target image | $v_2$ | Local crop |
| Mask probability | / | 0.5 |
| Mask ratio | / | $[0.1, 0.5]$ |
| High-resolution crop size | / | $(224 \times 224)$ |
| Low-resolution crop size | / | $(96 \times 96)$ |
| Global crop size | / | $(224 \times 224)$ |
| Local crop size | / | $(224 \times 224)$ |
| Optimizing settings | | |
| Optimizer | / | AdamW |
| Learning rate scheduler | / | Cosine |
| Weight decay | / | $0.04 \rightarrow 0.4$ |
| Momentum | / | $0.992 \rightarrow 1$ |
| Number of ViT encoder blocks | / | 12 |
| Patch size | $p$ | 16 |
| Base learning rate | $blr$ | $2 \times 10^{-3}$ |
| Decoder learning rate | / | $0.1 \times lr$ |
| Epochs | / | 50 |
| Warm-up epochs | / | 20 |
| Batch size | bs | 1024 |
| Number of ViT feature dim. | $d$ | 384 |
| Loss function | | |
| Weight of reconstruction loss | $\lambda_1$ | 0.8 |
| Weight of DINO loss | / | 1 |
| Weight of koleo loss | $\lambda_2$ | 0.1 |

Table A.1: The hyperparameters settings for our Crop-CoRe framework during pre-training.

## A.2 EVALUATION SETTINGS

In this section, we give details on how the methods are evaluated on the dense label propagation task. During evaluation, the pre-trained network $f$ is frozen. As initially introduced by Jabri et al. (2020), given a set of $T$ reference images $\boldsymbol{I}_r \in \mathbb{R}^{T \times H \times W \times 3}$, their dense one-hot labels $\boldsymbol{Y}_r \in \mathbb{R}^{T \times H \times W \times C}$, and a target image $\boldsymbol{I}_t \in \mathbb{R}^{H \times W \times 3}$, their dense representations are first computed with $f$ to form $\boldsymbol{X}_r = f(\boldsymbol{I}_r) \in \mathbb{R}^{T \times H \times W \times D}$ and $\boldsymbol{X}_t = f(\boldsymbol{I}_t) \in \mathbb{R}^{h \times w \times D}$, where $D$ is the latent dimension, and $C$ is the number of classes. Hence, to propagate their labels to the target, we proceed as follows:

$$\hat{\boldsymbol{Y}}_t = \arg\max(\text{Softmax}(\boldsymbol{X}_t \boldsymbol{X}_t^\top \odot \boldsymbol{M}/\tau, \, \text{dim} = -1)\boldsymbol{Y}_r, \, \text{dim} = -1) \tag{A.1}$$

where $\boldsymbol{M}$ is a mask giving the spatial region attended by each pixel, or patch, and $\tau$ is a temperature parameter. However, in practice, rather than aggregating from all the reference pixels, only the top

$K$ most similar are used. $T$ represents the length of the queue, and radius describes the visible region around each target pixel in the mask $M$. These hyperparameters are detailed in table A.2. These values follow (Eymaël et al., 2024) for fair comparison.

| Config | DAVIS-2017 | VIP | JHMDB |
|---|---|---|---|
| Top-K | 7 | 10 | 7 |
| Queue Length | 20 | 20 | 20 |
| Neighborhood Size | 20 | 20 | 20 |
| Temperature | 0.7 | 0.7 | 0.7 |

Table A.2: The hyperparameters settings for our T-CoRe framework during downstream evaluations.

