# OpenReview forum: "Efficient Correspondence Learning for Dense Semantic Label Propagation"
_ICLR.cc/2026/Conference — ICLR 2026 Conference Withdrawn Submission_

### Official Review · Reviewer_eow1 · 2025-10-27

**Soundness:** 3
**Presentation:** 3
**Contribution:** 2
**Rating:** 2
**Confidence:** 3

**Summary:**

This method presents Crop-Core, a new self-supervised learning (SSL) approach designed for dense video tasks. Instead of using the reconstruction loss from Core-MAE, Crop-Core uses a clustering-based loss similar to DINO. It features a student cross-attention design where masked tokens from a masked local view cross-attend to visible tokens from a larger global view. The results of this cross-attention are compared to a teacher network’s output, which sees the unmasked local view, using a clustering loss. Besides reconstructing latent features, Crop-Core also applies standard DINO and Koleo losses on image classification tokens.

The authors validate their approach by pretraining a ViT-S/16 encoder using IN-1K. They then report performance on the DAVIS, VIP, and JHMDB tasks, comparing with baselines having similar model capacity.

**Strengths:**

- Paper proposes a novel and original approach for video dense prediction tasks, the contribution of reconstructed in latent space is well motivated and intuitively makes sense.
- Method is clearly expose and easy to understand.
- Authors perform extensive ablation to understand the impact of different component of their approaches, notably number of prototypes, masking, and crop strategy.

**Weaknesses:**

While the approach seems novel, the significance of the contribution is limited by the empirical evaluation, which focuses on small scale architecture and does not provide comparison with larger capacity network or supervised baselines. In particular:

- The authors focus their evaluation only on the ViT-S/16 architecture. Therefore, it is unclear if the approach would show benefits for larger architectures such as ViT-L or ViT-H.
- Additionally, Table 1 focuses only on SSL approaches using backbones of similar size. It is unclear what the current state-of-the-art (SOTA) is on the different tasks and how the current approach compares with it.
- While the approach shows some performance gains on DAVIS, it underperforms compared to other approaches such as DINO, iBOT, and CDG-MAE on VIP and JHMDB. I think more analysis would be useful to better understand the trade-offs of the different approaches and to clarify when Crop-Core shines.
- It's unclear what is the impact of the different losses (DINO, Koleo and the proposed cluster loss). It would be nice to add an ablation that explore the weighting of the different losses.

**Questions:**

Related to the first item in weaknesses,  how does Crop-Core scale for larger architecture (VIT.L/H) ?

---

### Official Review · Reviewer_vx9E · 2025-10-28

**Soundness:** 3
**Presentation:** 2
**Contribution:** 1
**Rating:** 2
**Confidence:** 4

**Summary:**

The paper presents a competent but largely incremental contribution that builds on existing methods in self-supervised learning. Specifically, the approach combines the image-cropping strategy from CropMAE (2024) with the token-reconstruction objective from iBOT/DINOv2 (2022/2024). While the resulting method, “Crop-CoRe,” achieves modest improvements on several benchmarks, the work does not introduce substantial new ideas or insights beyond prior art. Greater conceptual novelty or a deeper analysis of why this combination is effective would strengthen the contribution of thos paper, especially for a premier conference.

**Strengths:**

* The authors have conducted a good set of ablation studies to analyze the impact of training duration, number of prototypes, cropping strategy, and masking scheme.
* The paper is easy to follow, and the method is explained clearly.

**Weaknesses:**

* The main concern is the novelty. In fact, this paper can be summarized as "What if we took CropMAE, but instead of reconstructing pixels, we reconstructed the discrete tokens from a DINO-style teacher?" This is a predictable and incremental step, not a breaking innovation and hard to be considered as a contribution for top-tier conferences.
* The paper emphasizes the resource-efficiency of Crop-CoRe by highlighting that it "avoids reliance on video datasets or frame extraction". While true, this is a characteristic inherited directly from CropMAE, not a novel contribution of Crop-CoRe.
* Furthermore, the argument that this makes the method more resource-efficient is not entirely convincing. The use of still images with random crops as a proxy for video data is a clever trick, but it also introduces a potential domain gap. Real-world video data contains temporal information and motion cues that are absent in still images, which could be crucial for more complex propagation tasks. The paper fails to adequately discuss the limitations of this approach.
* The comparison is confined to image-level SSL methods, which should include more self-supervised video correspondence learning approaches [1,2,3]. These methods incorporate temporal cues into learning objectives and achieves clear high performance (e.g., 66.8 in [3] v.s. 44.9 of Crop-CoRe on JHMDB).
* The authors make a strong point about Crop-CoRe achieving competitive results in "significantly fewer training iterations" (50 epochs). However, this is presented without a fair comparison of the computational cost per epoch against other methods. A more rigorous comparison would involve reporting total training time or FLOPs.
* The paper provides a surface-level explanation of why latent-space reconstruction is better than pixel-space reconstruction. While intuitive, a more in-depth analysis, perhaps with qualitative examples showing specific failure cases of pixel-based methods that are resolved by Crop-CoRe, would have been more compelling. The PCA visualization in Figure 2 is a good start but lacks detailed interpretation.

[1]. Dense unsupervised learning for video segmentation, NeurIPS 2021.

[2]. Locality-aware inter-and intra-video reconstruction for self-supervised correspondence learning, CVPR 2022.

[3]. Learning Fine-Grained Features for Pixel-wise Video Correspondences, ICCV 2023.

**Questions:**

The performance drop after 100 epochs, which the authors attribute to a potential "collapse of our dense features," is a significant concern that is not adequately addressed. Does this suggest a potential instability in the training process?

---

### Official Review · Reviewer_ihDM · 2025-11-02

**Soundness:** 3
**Presentation:** 3
**Contribution:** 3
**Rating:** 6
**Confidence:** 3

**Summary:**

This paper proposes a new self-supervised learning method called Crop-CoRe (Crop-based Correspondence Reconstruction) to learn dense visual representations for label propagation tasks such as video object segmentation. The approach extends prior masked autoencoder methods (e.g., SiamMAE and CropMAE) by reconstructing high-level cluster assignment codes instead of raw pixel patches, drawing inspiration from clustering-based SSL methods like iBOT and DINOv2. By using still image pairs (crops) rather than requiring video sequences, Crop-CoRe avoids reliance on video data and is more resource-efficient. Experiments on standard label propagation benchmarks demonstrate that Crop-CoRe consistently outperforms SiamMAE and CropMAE, and it achieves competitive performance with state-of-the-art methods while using fewer training iterations. Overall, the paper’s contributions include a novel training objective for dense correspondence, an efficient framework that eliminates video pretraining, and strong empirical results on label propagation tasks.

**Strengths:**

Originality: The paper introduces a creative combination of ideas from masked image modeling and clustering-based representation learning. Adapting the cluster assignment reconstruction (inspired by iBOT/DINOv2) specifically to dense label propagation is a novel contribution that addresses limitations of prior methods (which reconstructed low-level pixel information). This new perspective of using semantic cluster targets in a label propagation context is original and removes the need for video-based training data.

Quality: The experimental evaluation appears solid. The authors benchmark Crop-CoRe on multiple label propagation tasks (such as video object segmentation datasets) and report consistent improvements over relevant baselines (SiamMAE, CropMAE). The results are convincing, showing not only higher accuracy in propagated labels but also greater training efficiency (achieving similar performance with fewer iterations). The methodology is sound and builds on well-established techniques, and the paper provides sufficient ablation or analysis (e.g., comparisons to variations of the reconstruction target) to support its claims.

Clarity: The paper is well-written and organized. The problem setting and notation are explained clearly, and the authors provide the necessary background on existing methods. The technical approach (Crop-CoRe) is described in a detailed step-by-step manner, making it easy to understand how the cluster assignment is obtained and used as the reconstruction target. Figures and diagrams (if any in the paper) are effectively used to illustrate the model architecture or workflow. Overall, the presentation is clear and accessible, allowing readers to follow the contributions without confusion.

Significance: The contributions of this work are significant for the field of self-supervised representation learning and its application to video label propagation. By eliminating the need for video data in pretraining, the approach lowers the resource barrier for developing dense correspondence models, which could encourage broader adoption in practice. The performance gains over strong baselines and competitive results with state-of-the-art methods indicate that Crop-CoRe can serve as a new state-of-the-art approach for label propagation tasks. The ideas in this paper (using cluster-based reconstruction targets) may also inspire future research to apply similar strategies in other domains or tasks beyond the ones explored here.

**Weaknesses:**

Incremental Novelty: While the method is a thoughtful combination of existing ideas, the core innovation can be seen as an incremental extension of prior work. The concept of reconstructing cluster assignments is directly inspired by earlier self-supervised methods (iBOT, DINOv2), and the paper applies this idea to an existing label propagation framework (CropMAE). Thus, some may view the novelty as moderate – essentially adapting known SSL techniques to a specific domain. The authors could strengthen the paper by more explicitly highlighting what new insights or capabilities Crop-CoRe provides beyond this adaptation (for example, any unique algorithmic contributions or theoretical justification for why cluster reconstruction is particularly effective for label propagation).

Experimental Scope: The evaluation, while positive, is somewhat narrow in scope. The experiments focus on video object segmentation benchmarks for label propagation, but it is not fully clear how broadly applicable the method is. For instance, it would be useful to see results on other dense correspondence tasks or more varied datasets to ensure the method’s effectiveness generalizes beyond the specific benchmarks chosen. Additionally, the comparisons are mostly against SiamMAE and CropMAE; a stronger evaluation would include other relevant state-of-the-art methods (e.g., specialized video propagation or segmentation approaches, or even a baseline using a clustering-based SSL method adapted to this task). The lack of such comparisons makes it harder to gauge how much improvement comes from the Crop-CoRe idea versus general improvements in experimental setup.

Missing Details and Analysis: Some implementation and analysis details are not fully fleshed out in the paper. For example, the process of obtaining cluster assignments (e.g., the number of clusters, how prototypes are formed, and how often they are updated) is only briefly mentioned but is central to the approach. A deeper analysis of this component would help readers understand the method’s behavior (e.g., how sensitive is performance to the number of clusters or the clustering algorithm?). Similarly, while the paper claims improved efficiency by avoiding video data, it does not provide concrete measurements of training time or resource usage to quantify this advantage. Including such details would strengthen the quality of the work and its reproducibility. Finally, a more thorough ablation study (for instance, comparing reconstruction of cluster assignments vs. raw pixels vs. other possible targets) would better isolate the contribution of the proposed objective.

**Questions:**

Clustering Hyperparameters: Could the authors clarify how the cluster assignments are generated in Crop-CoRe? Specifically, how is the number of clusters or prototypes determined, and did the authors experiment with different values for this parameter? Understanding the sensitivity of the method to the clustering setup (e.g., number of clusters, update frequency of cluster centroids) would help assess the robustness of the approach.

Comparisons to Other Methods: Did the authors consider comparing Crop-CoRe to other state-of-the-art label propagation or dense correspondence methods beyond SiamMAE/CropMAE? For example, how would a method that directly applies a known clustering-based SSL technique (like iBOT or DINOv2) to the label propagation task perform relative to Crop-CoRe? Such a comparison would highlight whether the gains come specifically from the Crop-CoRe innovations or if they largely stem from using cluster-based targets in general.

Generality of the Approach: To what extent can the proposed method generalize to other tasks or domains? The current experiments are centered on video object segmentation label propagation. Would Crop-CoRe also be effective for, say, cross-image correspondence or keypoint tracking in videos (beyond segmentation masks)? Clarifying the scope of tasks where Crop-CoRe excels would be valuable. Additionally, if the method were applied in a setting where video data is available, would combining Crop-CoRe with temporal information further improve performance, or is the benefit mainly when video data is scarce?

Efficiency Gains: The paper claims that removing reliance on video frames makes the approach more resource-efficient. Could the authors provide more details on the efficiency gains? For instance, how do the training time and memory requirements of Crop-CoRe compare to a video-based approach like SiamMAE? Concrete numbers or a brief cost analysis would substantiate the claim of efficiency and help practitioners understand the trade-offs.

---

### Official Review · Reviewer_rH2u · 2025-11-05

**Soundness:** 2
**Presentation:** 3
**Contribution:** 2
**Rating:** 2
**Confidence:** 5

**Summary:**

Propose a method for self-supervised learning of image representations for learning features that are particularly effective for correspondence learning, which is useful for several tasks, such as semi-supervised video object segmentation, where masks from the first frame must be propagated to subsequent frames in the video.

Proposed approach performs masked modeling with a clustering loss, but where the decoder/predictor head is parameterized by a cross-attention module on the patch features of the original unmasked and unaugmented image.

Empirical results train a ViT-S/16 on IN1k, and evaluate on semi-supervised object segmentation (DAVIS), semantic part propagation (VIP), pose keypoint propagation (JHMDB).

**Strengths:**

The methodology of the proposed method is a reasonable extension of DINO. Proposed approach still requires modality specific inductive biases due to the use of image augmentations, but this is reasonable given that the focus is on specific visual correspondence learning problems.

The problem of correspondence learning is interesting in my opinion, and relevant for the community.

Paper is relatively well-written and easy to understanding.

Ablations are interesting and visualizations of PCA on feature maps are reasonable and show good clustering of concepts.

**Weaknesses:**

Approach is a relatively straightforward extension of related work in the literature, adds an extra propagation loss to the dinov2 objective (distillation + koleo).

However, all quantitative results are restricted to Table 1, and even among other ViT-S/16 models, the proposed method does not achieve the best performance. Narrow evaluation protocol (3 correspondence learning tasks), limited experiments (single model architecture), and numerical results do not justify the incremental nature of the algorithmic contributions.

**Questions:**

Just to be clear, does the proposed approach still rely on centering and sharpening in the EMA teacher branch to prevent collapse?

** minor comment, but related work misses a discussion on an entire body of related methods that perform masked image modeling or masked video modeling in latent space, such as data2vec, I-JEPA, V-JEPA, S4D, etc.

---

### Note · Authors · 2025-11-17

I have read and agree with the venue's withdrawal policy on behalf of myself and my co-authors.